# Effects of dexmedetomidine as a perineural adjuvant for femoral nerve block: A systematic review and meta-analysis

**Zi-Fang Zhao[1◉], Lei Du[2◉], Dong-Xin Wang[1]***

**1** Department of Anesthesiology and Critical Care Medicine, Peking University First Hospital, Beijing, China,
**2** Department of Radiology, China-Japan Friendship Hospital, Beijing, China

◉ These authors contributed equally to this work.
* wangdongxin@hotmail.com, dxwang65@bjmu.edu.cn

## Abstract

**Data Availability Statement:** All relevant data are within the manuscript and its Supporting Information files.

**Funding:** The authors received no specific funding for this work.

### Background

Femoral nerve block (FNB) is one of the first-line analgesic methods for patients following lower extremity surgery. However, FNB with local anesthetics alone exert limited potency and supplemental opioids are often required. Dexmedetomidine (DEX) has been used to improve the analgesic effects of FNB. The present systematic review and meta-analysis were conducted to evaluate the effectiveness of DEX as an adjuvant to local anesthetics for FNB.

### Methods

Randomized controlled trials comparing the effects of DEX versus sham control in combination with local anesthetics for FNB were included in this meta-analysis. Postoperative pain scores, duration of analgesic effects, and postoperative narcotic consumption were outcomes of interest. This research was performed according to the Preferred Reporting Items for Systematic Reviews and Meta-Analyses (PRISMA) statements.

### Results

A total of 9 studies encompassing 580 participants were included for data synthesis after critical evaluation. DEX as an adjuvant with local anesthetics for FNB significantly relieved pain intensity at 12, 24 and 48 hours after surgery, both at rest (standardized mean difference -1.34 [95% CI -1.87 to -0.82], P<0.00001 at 12 h; -1.26 [-1.90 to -0.0.63], P<0.0001 at 24 h; and -1.34; [-2.18 to -0.50], P = 0.002 at 48 h) and with movement (-1.30 [-2.17 to -0.43], P = 0.004 at 12 h; -1.02 [-1.31 to -0.72], P<0.00001 at 24h; and -1.33 [-2.03 to -0.63], P = 0.0002); it also significantly prolonged analgesic duration (mean difference 7.23 h [95% CI 4.07 to 10.39], P<0.00001) and decreased opioid consumption (mean difference of morphine equivalent -12.13 mg [95% CI -23.36 to -0.89], P<0.00001). Regarding safety, DEX use increased the rate of hypotension (odds ratio 4.10, 95% CI 1.40 to 12.01, P = 0.01).

**Competing interests:** The authors have declared that no competing interests exist.

## Conclusion

DEX as an adjuvant to local anesthetics for FNB improves analgesia, prolongs analgesic duration and reduces supplemental opioid consumption; but increases hypotension.

## Introduction

As an easily operable and conventional technique, femoral nerve block (FNB) remains one of the first-line analgesic options for acute pain following lower extremity surgeries such as knee arthroplasty, femoral shaft fractures, knee arthroscopy, and cruciate ligament reconstruction [1, 2]. With reliable and effective analgesia, FNB decreases opioid consumption, minimizes opioid-related side effects, accelerates postoperative recovery and improves quality-of-life [3, 4]. However, when used for FNB, local anesthetics alone often exert limited potency of analgesia and are insufficient to avoid supplemental opioid usage. Whereas increasing the dose or concentration of local anesthetics for FNB may increase the risks of toxic effects and motor block, the latter may adversely affect quadriceps strength and postpone early off-bed ambulation [5–7]. Consequently, many adjuvants to local anesthetics, such as epinephrine, clonidine and glucocorticoids, are investigated extensively in order to prolong pain relief [8, 9].

Dexmedetomidine (DEX), a highly selective and potent α2-adrenergic receptor agonist, is widely used in clinical settings due to its properties of sedation, anxiolysis, analgesia, and sleep promotion [10–14]. Apart from the authorized intravenous infusion regimen, DEX has been increasingly employed to intensify the analgesic effects of nerve blocks, which is an off-label indication. Recent studies and meta-analyses indicate that DEX possesses favorable effects in prolonging the duration of peripheral nerve block, improving the efficacy of pain relief and reducing narcotic consumption [15–18]. Accordingly, growing evidence have elucidated the anti-inflammatory, sleep-promoting and supplemental analgesic effects of adding DEX to local anesthetics for FNB [19–27]. Considering these emerging studies, we carried out the present systematic review and meta-analysis of randomized controlled trials (RCTs) to evaluate the benefit and effectiveness of DEX as adjuvants to local anesthetics for FNB.

## Methods

This systematic review and meta-analysis of randomized controlled trials was performed according to the Preferred Reporting Items for Systematic Reviews and Meta-Analyses (PRISMA) statement and the Cochrane Collaboration.

International databases (PubMed, EMBASE, Cochrane Library, and Web of Science) were searched by two authors (Zi-Fang Zhao and Lei Du) independently from the inception to February 2020. To avoid omitting the potentially relevant articles, we used the Medical Subject Headings (MeSH) terms and corresponding free text words: "dexmedetomidine" (MeSH term) OR "Precedex" OR "medetomidine", then combined with "femoral nerve block" by the Boolean operator "AND". All terms were searched in the Title, Abstract, and Keywords sections. Subsequently, the identified articles were screened by reading the title and retrieved abstracts. Full text of selected articles was carefully read for possible inclusion. We also checked the reference lists of selected articles to avoid the omission of any eligible trials. There was no restriction regarding the publication language.

## Inclusion and exclusion criteria

Studies included in this meta-analysis should meet the following criteria: (a) participants received FNB for postoperative multimodal analgesia; (b) compared the effects of DEX versus sham control used in combination with local anesthetics for FNB; (c) reported at least one of the following predesigned outcomes: postoperative pain scores, duration of analgesic effects or postoperative narcotic consumption; (d) study design: RCTs.

Studies were excluded if they met any of the following criteria: (a) non-RCT studies; (b) abstracts presented at meetings, reviews, letters, case reports or editorials; (c) animal studies; (d) analgesic effect not assessed. Any disagreements regarding study selection were resolved by group discussion and consensus.

## Data extraction and outcome assessment

Two reviewers (Zi-Fang Zhao and Lei Du) extracted important variables from the included studies independently and recorded them in a predesigned database. Any discrepancy during information extraction was reevaluated more seriously and decided by discussion. The following information was collected from each article: first author; year of publication; study design; geographical location; sample size; participant characteristics, including mean age, gender distribution and type of surgery; inclusion and exclusion criteria; type and duration of surgery; primary and secondary endpoints; results and statistical data.

Mean and standardized deviation (SD) were used to describe the extracted data. When median and range or interquartile range (IQR) were presented, the mean and SD were estimated by using the equation introduced in the Cochrane Handbook for Systematic Reviews of Interventions [28] and previous papers [10, 29, 30]. We only extracted data of the DEX group and the sham control group if the studies divided subjects into several interventional arms and used other anesthetics. When multiple DEX regimens were used, we extracted only data from the group which accepted the highest dose [28]. For studies that reported missing or insufficient data, we attempted to contact authors to acquire data. When feasible, data were extrapolated from figures or tables. The software GetData Graph Digitizer (v2.26, Canopus, Japan) was used to extract the outcome values when the results were reported as a graph.

## Risk-of-bias assessment

To determine the methodological quality of included studies, two authors independently performed the evaluation [31]. Seven domains were covered when rating the risk of bias: random sequence generation, allocation concealment, blinding, outcome assessment, incomplete outcome data, selective reporting, and other bias. The risk of bias for each domain was judged as "high," "low," or "unclear".

## Statistical analysis

Review Manager software (RevMan, version 5.3, Cochrane Library, Oxford, UK) was used to perform data synthesis and statistical analysis. Considering that postoperative pain was evaluated with either the visual analogue scale (VAS) or numerical rating scale (NRS), both scores were converted to a standardized 0–10 scale and standardized mean difference (SMD) was calculated for quantitative synthesis. Whereas the weighted mean difference (WMD) was used to evaluate the narcotic consumption and the duration of analgesic efficacy. Different opioids were converted into morphine equivalents for synthesizing data of narcotic consumption. Odds ratio (OR) with 95% confidence interval (CI) was used to assess the dichotomous data (incidences of adverse effects) when applicable. Forest plots were used to present the pooled

results and corresponding 95% CIs. Cochrane Q test ($P<0.10$ for a statistical significance) and I-square ($I^2$) test were performed to evaluate the heterogeneity among included studies. As described in the Cochrane review guidelines, $I^2 >50\%$ indicated a significantly high heterogeneity and the corresponding outcome variables were analyzed with the random effect model [32]. For $I^2 <50\%$, either random or fixed effect model was appropriate. We analyzed and looked for possible underlying sources of heterogeneity for the included trials, and identified the clinical, methodological or statistical variations (severity of illness, administration regimen of DEX, type of surgery, multimodal analgesia protocol, and etc.). Identified heterogeneities were resolved with subgroup analysis when two or more studies were included in each subgroup. In addition, according to the results of quality evaluation, we performed a sensitivity analysis by excluding the article with a significantly high risk of bias. After the meta-analysis of each included analgesic outcome, the quality of evidence was evaluated with the Grading of Recommendations Assessment, Development and Evaluation (GRADE) system. Based on assessment results in five aspects (risk of bias, inconsistency, indirectness, imprecision, and publication bias), evidence was graded as high, moderate, low, or very low. A $P$ value of $<0.05$ was considered statistically significant.

## Results

### Literature search

A total of 157 related records were identified during the initial literature search conducted in February 2020. After careful checking and removing duplications, the abstracts of the remaining 91 records were carefully read. The full texts of 12 articles [19–27, 33–35] were acquired from the electronic databases and assessed for the possibility of inclusion. One single-blind trial was excluded because the researchers administered DEX without local anesthetics for peripheral nerve block [35]. Two trials failed to compare the effects of DEX with sham control [33, 34]. Ultimately, 9 RCTs [19–27] with 580 participants were included for data synthesis after critical assessment (Fig 1).

### Study characteristics

Table 1 presented the main clinical features of 9 eligible studies (included 580 patients) for qualitative and quantitative synthesis of efficacy and safety data. All these studies were randomized, double-blinded trials designed to investigate the analgesic effects of DEX combined with local anesthetics for FNB and compared with sham control. Types of surgery included arthroscopic knee surgery [19, 22] and total knee arthroplasty [20, 21, 23–27]. DEX were used for single-shot FNB in 5 studies [19–23] and for continuous FNB in 4 studies [24–27]. DEX were administered at a constant dosage [19, 22], a constant infusion rate [26, 27], or a dosage according to patients' body weight [20, 21, 23–25].

### Risk of bias

Eight studies [19–23, 25–27] clearly described the generation methods of random sequences and, in 4 of them [21, 22, 25, 26], the random sequences were sealed in opaque envelopes. The implementation of blinding for participants and study personnel were present in 6 studies [19, 22–26]; blinded outcome assessment was performed in 5 studies [19, 22, 24–26]. One study reported incomplete data due to loss to follow-up caused by accidental dislocation of the catheter in some patients [27]. More details were presented in Fig 2.

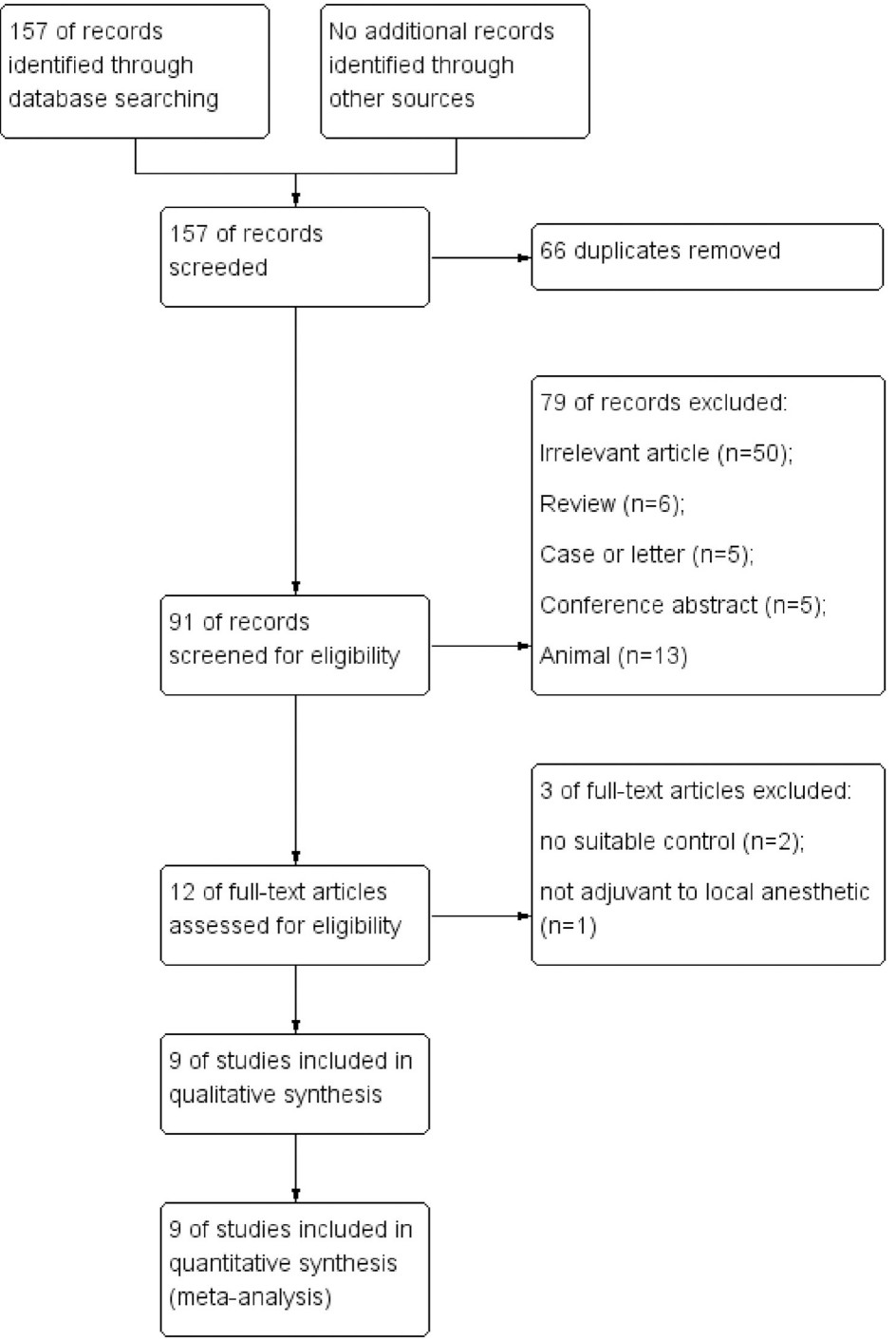

**Fig 1. Flow diagram showing literature search results.**

## Postoperative pain intensity

Eight studies evaluated the VAS or NRS pain scores but data were accessible in 6 studies for quantitative analysis [19, 21, 23, 25–27]. We analyzed the pain scores at 12, 24 and 48 hours after surgery, respectively, in the resting and active state.

**Table 1. Study characteristics of all randomized trials included in the meta-analysis.**

| Study | Country | Sample size | | Surgery | Anesthesia | FNB protocol | DEX interventions | Local anesthetics |
|---|---|---|---|---|---|---|---|---|
| | | DEX (male) | Control (male) | | | | | |
| Abdulatif 2016 [22] | Egypt | D1, 15 (15); D2, 15 (15); D3, 15 (12) | 15 (13) | arthroscopic knee surgery | GA | single-shot | D1, 25 μg; D2, 50 μg; D3, 75 μg | 0.5% bupivacaine |
| Deng 2018 [19] | China | 30 (24) | 30 (20) | arthroscopic knee surgery | CSEA | single-shot | 100 μg | 0.25% ropivacaine |
| Li 2017 [21] | China | 30 (20) | 30 (20) | TKA | GA | single-shot | 1 μg/kg | 0.5% ropivacaine |
| Packiasabapathy 2017 [23] | India | D1, 20 (7); D2, 20 (8) | 20 (6) | TKA | SA | single-shot | D1, 1 μg/kg; D2, 2 μg/kg | 0.25% bupivacaine |
| Pan 2017 [20] | China | 30 (17) | 30 (16) | unilateral TKA | GA | single-shot | 1 μg/kg | 0.25% ropivacaine |
| Sharma 2016 [24] | India | 25 (8) | 25 (14) | unilateral TKA | SA | continuous | 1.5 μg/kg | 0.2% ropivacaine |
| Wang 2018 [27] | China | 80 (20) | 80 (17) | single TKA | SA | continuous | 0.1 μg/kg/h | 0.2% ropivacaine |
| Yang 2019 [25] | China | 30 | 30 | TKA | GA | continuous | 2 μg/kg | 0.1% ropivacaine |
| Zhao 2019 [26] | China | D1, 30 (14); D2, 30 (10) | 30 (18) | TKA | SA | continuous | D1, 0.2 μg/kg/h; D2, 0.5 μg/kg/h | 0.15% ropivacaine |

**Abbreviations**: DEX, dexmedetomidine; D, dexmedetomidine intervention groups; TKA, total knee arthroplasty; GA, general anesthesia; CSEA, combined spinal-epidural anesthesia; SA, spinal anesthesia; FNB, femoral nerve block.

**Pain scores in resting state.** The postsurgical resting pain scores were reported at 12 hours in 5 studies including 280 patients [19, 21, 23, 25, 26] and at 24 and 48 hours in 6 studies including 440 patients [19, 21, 23, 25–27]. DEX was used as an adjuvant for single-shot FNB in 3 studies [19, 21, 23] and for continuous FNB in 3 studies [25–27]. The combined data showed a significant difference between DEX and control groups in the resting pain score at 12 hours after surgery (SMD = -1.34; 95% CI = -1.87 to -0.82; $P<0.00001$; $I^2$ = 74%). Significantly reduced pain scores at rest were also found in DEX-treated patients at 24 hours (SMD = -1.26; 95% CI = -1.90 to -0.63; $P<0.0001$; $I^2$ = 88%) and 48 hours (SMD = -1.34; 95% CI = -2.18 to -0.50; $P$ = 0.002; $I^2$ = 93%) following surgery. Sensitivity analysis was conducted to detect the origin of heterogeneity, but no notable changes were found in all three timepoints after excluding any of these studies. The current results indicated that DEX added to local anesthetics for FNB significantly lowered postoperative pain intensity in resting state (Fig 3).

**Pain scores in active state.** A total of 5 studies including 280 participants [19, 21, 23, 25, 26] assessed pain scores in active state at 12 postoperative hours; 6 studies including 440 participants [19, 21, 23, 25–27] reported pain score in active state at 24 and 48 postoperative hours. The pooled data showed that DEX added to local anesthetics for FNB significantly decreased pain scores in active state at 12 hours (SMD = -1.30; 95% CI = -2.17 to -0.43; $P$ = 0.004; $I^2$ = 91%), 24 hours (SMD = -1.02; 95% CI = -1.31 to -0.72; $P<0.00001$; $I^2$ = 49%) and 48 hours (SMD = -1.33; 95% CI = -2.03 to -0.63; $P$ = 0.0002; $I^2$ = 90%) after surgery. The effect of DEX in reducing active pain score was not altered after divided into single-shot FNB [19, 21, 23] and continuous FNB subgroups [25–27]. The present data indicated that DEX added to local anesthetics for FNB significantly lowered postoperative pain intensity in active state (Fig 4).

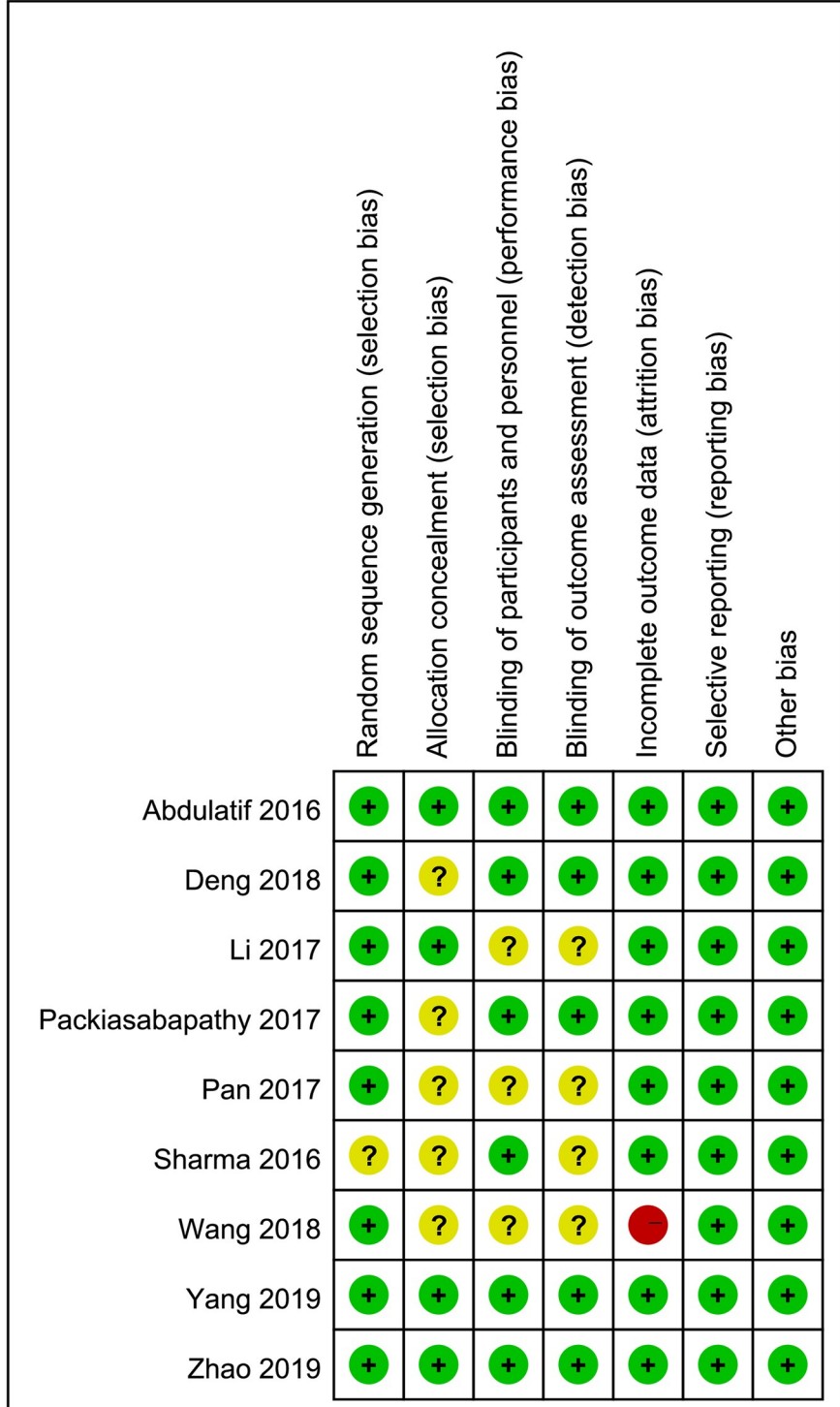

**Fig 2. Risk-of-bias evaluation for all included trials.**

## Duration of analgesic effects

Impacts of DEX on the analgesic durations of single-shot FNB were investigated in 5 trials including 240 patients [19, 20, 22–24]. The analgesic duration was defined as the time interval

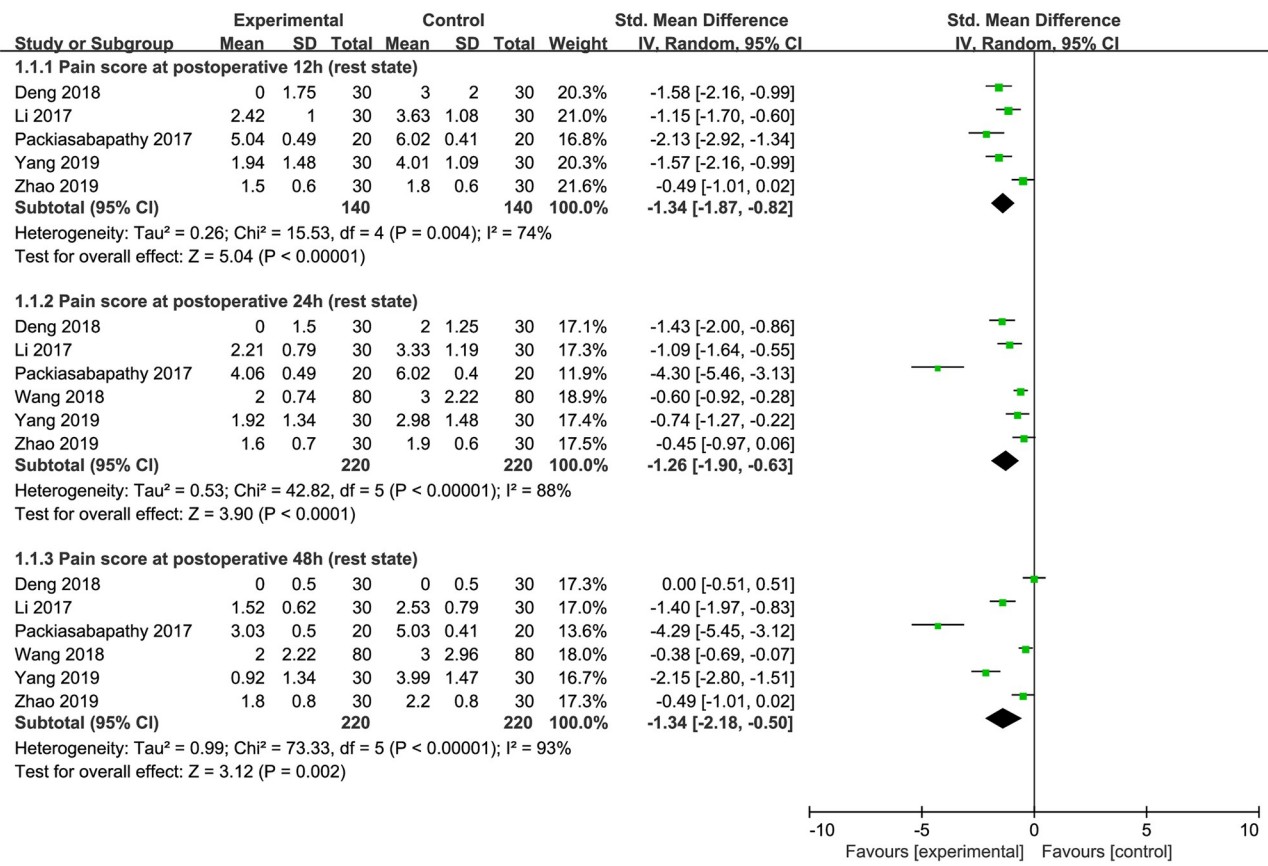

**Fig 3. Effects of dexmedetomidine versus placebo on the postoperative pain scores in resting state.**

from the conduct of FNB until the first use of patient-controlled analgesia (opioid or ropiva-caine) [20, 23, 24], the first demand of rescue morphine [22], or a patient complaint of NRS pain score of ≥4 [19]. Pooled results from these studies indicated a statistically significant pro-longation of analgesic duration in the DEX intervention groups (mean difference [MD] = 7.23 hours; 95% CI = 4.07 to 10.39; $P < 0.00001$; $I^2 = 96\%$; Fig 5). Sensitivity analysis was performed, but the recalculated MD and heterogeneity after excluding any of the trials showed no signifi-cant changes.

## Morphine equivalent consumption

A total of 4 studies [20, 22–24] explored opioid consumption at 24 hours after surgery. One study was excluded because of reporting incomplete data and the effects could not be esti-mated with RevMan software [24]. The combined results showed that morphine equivalent consumption was significantly decreased in patients who received FNB with DEX-local anes-thetic mixture (MD = -12.13 mg; 95% CI = -23.36 to -0.89; $P < 0.00001$; $I^2 = 97\%$; Fig 6).

## Adverse effects

Incidence rates of bradycardia [20, 22, 23], hypotension [20, 22, 23, 27] and postoperative nau-sea and vomiting (PONV) [26, 27] were pooled for analysis. The combined results demon-strated that DEX in combination with local anesthetics for FNB increased the risk of hypotension (OR = 4.10; 95% CI = 1.40 to 12.01; $P = 0.01$; $I^2 = 8\%$), but had no significant

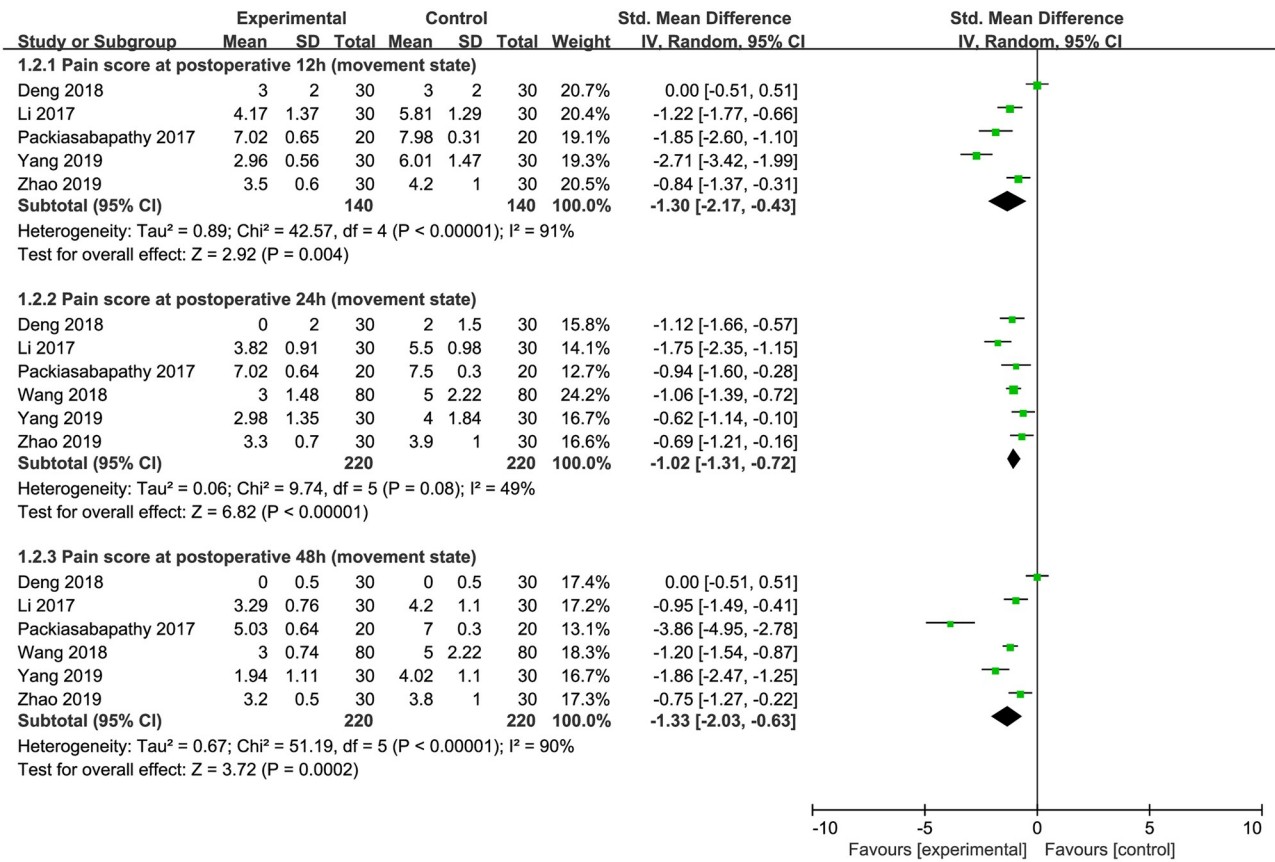

**Fig 4. Forest plots of the effects of dexmedetomidine versus placebo on the postoperative pain scores in active state.**

influence on the incidences of bradycardia (OR = 1.62; 95% CI = 0.14 to 18.67; $P$ = 0.70; $I^2$ = 63%); on the other hand, it tended to reduce PONV although not significantly so (OR = 0.36; 95% CI = 0.12 to 1.06; $P$ = 0.06; $I^2$ = 0%) (Fig 7). Sharma et al. [24] also reported that patients in the DEX group had significantly lower systolic blood pressure and mean arterial pressure during the early postoperative period.

## GRADE evidence

Details regarding the GRADE evidence evaluation are shown in Table 2. The level of evidence was moderate for resting pain score at postoperative 12 hours and active pain score at

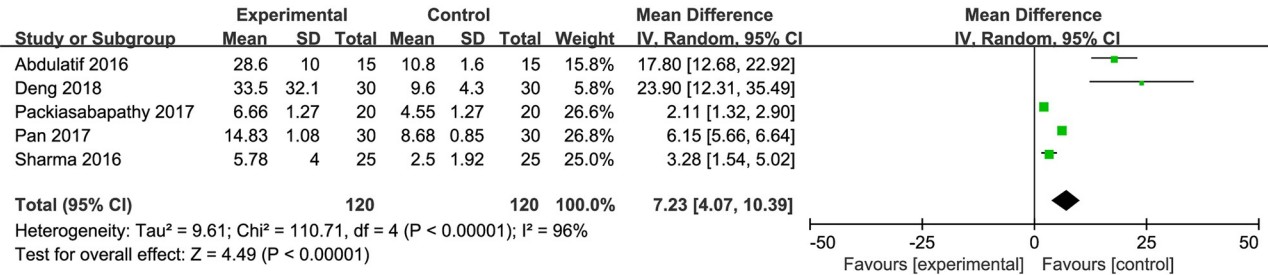

**Fig 5. Dexmedetomidine versus placebo on the analgesic duration of femoral nerve block.**

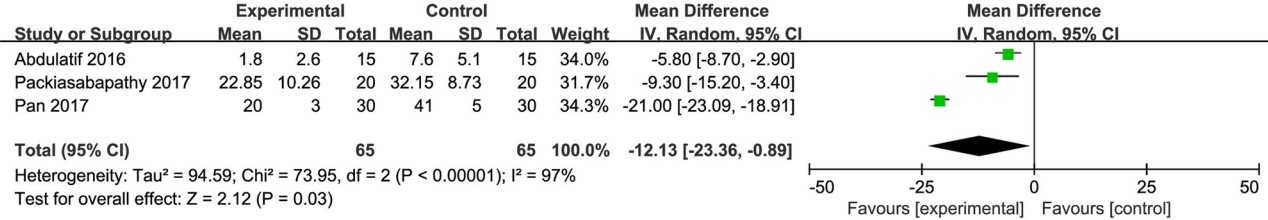

**Fig 6. Dexmedetomidine versus placebo on the postoperative consumption of morphine-equivalents.**

postoperative 24 hours; low for resting pain score at postoperative 24 hours, active pain score at postoperative 12 hours, duration of analgesic effects, and morphine equivalent consumption; very low for resting and active pain scores at postoperative 48 hours, and incidence of hypotension.

## Discussion

In the present systematic review and meta-analysis, we included 9 eligible RCTs to specifically evaluate the efficacy and safety of DEX combined with local anesthetics for FNB. Pooled results of the available data showed that perineural DEX combined with local anesthetics significantly improved analgesia both at rest and active state for up to 48 hours after surgery. In addition, combined use of DEX with local anesthetics for FNB significantly prolonged the duration of analgesia and reduced the cumulative consumption of rescue opioids. However, the increased risk of hypotension should be taken into consideration.

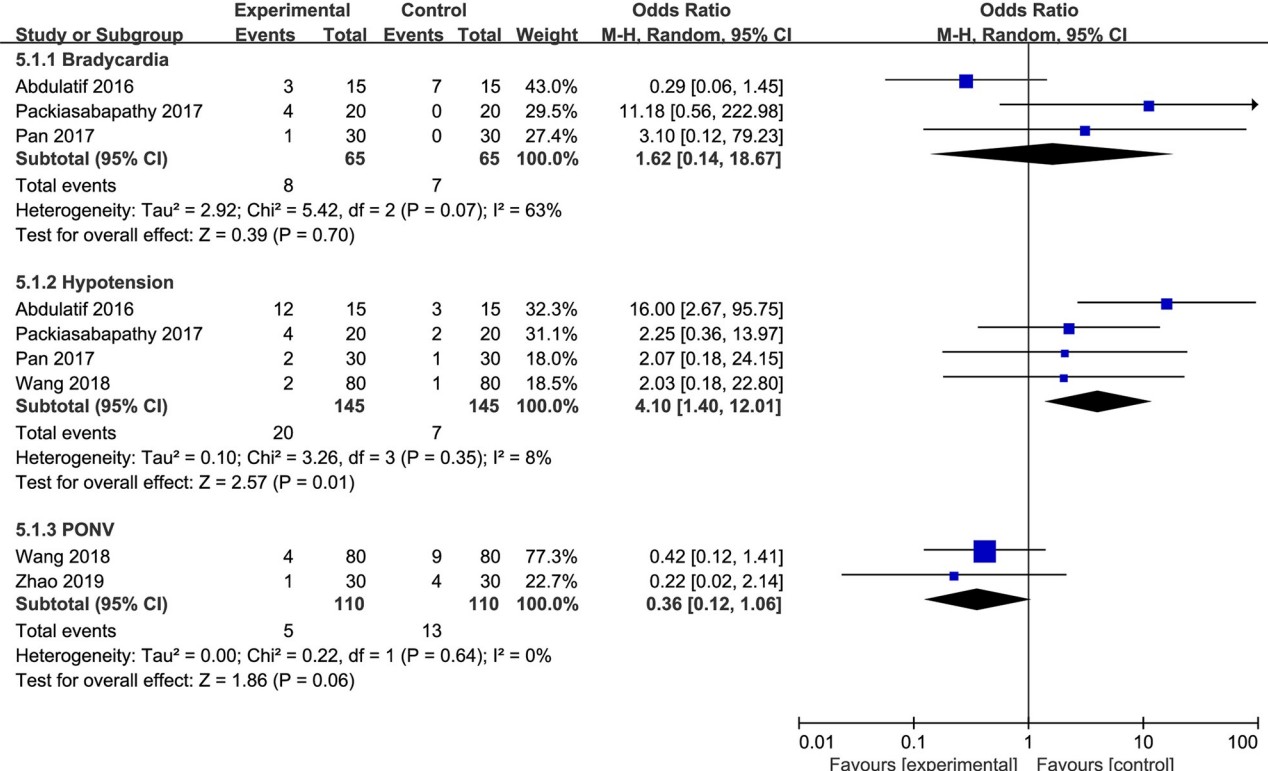

**Fig 7. Incidence rates of adverse effects.** PONV indicates postoperative nausea and vomiting.

**Table 2. GRADE evidence profile.**

| No. of studies | Study design | Quality assessment | | | | | No. of patients | | Effect | | Quality |
|---|---|---|---|---|---|---|---|---|---|---|---|
| | | Risk of bias | Inconsistency | Indirectness | Imprecision | Other | DEX | Control | Relative (95% CI) | Absolute (95% CI) | |
| Resting pain score at 12 hours after surgery | | | | | | | | | | | |
| 5 | RCT | not serious | serious [1] | not serious | not serious | none | 140 | 140 | — | SMD = -1.34 (-1.87 to -0.82) | ⊕⊕⊕◯ MODERATE |
| Resting pain score at 24 hours after surgery | | | | | | | | | | | |
| 6 | RCT | serious [2] | serious [3] | not serious | not serious | none | 220 | 220 | — | SMD = -1.26 (-1.90 to -0.63) | ⊕⊕◯◯ LOW |
| Resting pain score at 48 hours after surgery | | | | | | | | | | | |
| 6 | RCT | serious [2] | very serious [4] | not serious | not serious | none | 220 | 220 | — | SMD = -1.34 (-2.18 to -0.50) | ⊕◯◯◯ VERY LOW |
| Active pain score at 12 hours after surgery | | | | | | | | | | | |
| 5 | RCT | not serious | very serious [5] | not serious | not serious | none | 140 | 140 | — | SMD = -1.30 (-2.17 to -0.43) | ⊕⊕◯◯ LOW |
| Active pain score at 24 hours after surgery | | | | | | | | | | | |
| 6 | RCT | serious [2] | not serious | not serious | not serious | none | 220 | 220 | — | SMD = -1.02 (-1.31 to -0.72) | ⊕⊕⊕◯ MODERATE |
| Active pain score at 48 hours after surgery | | | | | | | | | | | |
| 6 | RCT | serious [2] | very serious [6] | not serious | not serious | none | 220 | 220 | — | SMD = -1.33 (-2.03 to -0.63) | ⊕◯◯◯ VERY LOW |
| Duration of analgesic effects | | | | | | | | | | | |
| 5 | RCT | not serious | very serious [7] | not serious | not serious | none | 120 | 120 | — | MD = 7.23 h (4.07 to 10.39) | ⊕⊕◯◯ LOW |
| Morphine equivalent consumption | | | | | | | | | | | |
| 3 | RCT | not serious | very serious [8] | not serious | not serious | none | 65 | 65 | — | MD = -12.13 mg (-23.36 to -0.89) | ⊕⊕◯◯ LOW |
| Hypotension | | | | | | | | | | | |
| 4 | RCT | serious [2] | serious [9] | not serious | serious [10] | none | 145 | 145 | OR = 4.10 (1.40 to 12.01) | — | ⊕◯◯◯ VERY LOW |

**Abbreviations**: DEX, dexmedetomidine; CI, confidence interval; RCT, randomized controlled trial; SMD, standardized mean difference; MD, mean difference; OR, odds ratio.

[1] Heterogeneity: $I^2 = 74\%$.

[2] One trial reported incomplete outcome data.

[3] Heterogeneity: $I^2 = 88\%$.

[4] Heterogeneity: $I^2 = 93\%$.

[5] Heterogeneity: $I^2 = 91\%$.

[6] Heterogeneity: $I^2 = 90\%$.

[7] Heterogeneity: $I^2 = 92\%$.

[8] Heterogeneity: $I^2 = 97\%$.

[9] Included different conclusions.

[10] The 95% CI was broad.

Despite the advances in surgical techniques and perioperative care, postoperative pain remains one of the most challenging problems for patients and physicians. More than 60% of hospitalized surgical patients experience moderate to severe postoperative pain, which may persist for up to 2 weeks after surgery [36]. Sufficient control of the acute pain helps to decrease patient anxiety, inhibit excessive stress response, shorten hospital stay, and facilitate

rehabilitation [37]. For a long time, opioids play a critical role in postoperative analgesia. But undesired adverse effects, such as respiratory depression, PONV, pruritus, gastrointestinal discomfort, and potential drug addiction, impede early physical rehabilitation and even long-term recovery. Various opioid-sparing analgesic approaches, including non-steroidal anti-inflammation drugs, intrathecal opioids, peripheral nerve block and multimodal analgesia, have been extensively investigated. By combining a variety of analgesic medications and techniques in order to reduce corresponding adverse effects, multimodal perioperative analgesia has been an indispensable component of Enhanced Recovery After Surgery [38].

FNB, which covers the anteromedial aspect of the knee and hip, is an effective analgesic technique for numerous lower extremity surgeries including knee arthroplasty, knee or hip arthroscopic surgery, and cruciate ligament reconstruction [39, 40]. However, FNB with local anesthetics alone has limited analgesic effect and duration; supplemental analgesics such as opioids are usually required. With these considerations in mind, great efforts have been made to find strategies that can improve the analgesic potency and prolong the analgesic duration of FNB. The current meta-analysis provides more reliable evidence regarding the applications of DEX as an adjuvant to local anesthetics (mostly ropivacaine or levobupivacaine) in FNB for postoperative analgesia.

Perineural DEX remains an off-label use, but the efficacy and safety have been verified repetitively in either pediatric or adult patients receiving brachial plexus block, thoracic para-vertebral block, transversus abdominis plane block, epidural analgesia, and caudal block. Synthesized results from these studies reached a consensus that DEX in combination with local anesthetics provided better analgesia than the local anesthetics alone, as evidenced by significantly lowered VAS/NRS pain scores, longer analgesic duration, and reduced opioids consumption [15, 16, 18, 41–44]; which are in line with our current results.

Several pharmacological and molecular mechanisms are speculated to contribute to the analgesia-promotion property of DEX in peripheral nerve block. With a higher affinity to the spinal and peripheral α2 adrenal receptors, DEX may produce synergic analgesic effects by suppressing the action potentials in the peripheral nerve fibers [45]. It has been proved that systematically administered DEX blunts surgery-related stress and inflammatory reactions, which may provide advantage in decreasing postoperative complications [10]. Interestingly, DEX combined with ropivacaine for FNB also significantly reduced the local concentrations of inflammatory cytokines in knee joint fluid when compared to sham control [21]. This may contribute to improved analgesia after surgery. Furthermore, both systemic and local administrations of DEX are able to improve postoperative sleep quality, whereas sleep disorders aggregate the intensity of postoperative pain [11, 27, 46]. The sleep-promotion features may also partly explain the pain relief effects of DEX utilized for FNB.

The favorable analgesic effects of DEX are found both in single-injection FNB and continuous FNB. It remains controversial whether the analgesia-promoting effect of continuous regimen is better than single-shot one. One meta-analysis [47] showed no significant difference between the two methods; in another article, however, more effective analgesia with continuous FNB was identified [48]. We attempted to separately analyze the effects of DEX as adjuvants in these two regimens, but found no notable changes in the postoperative pain scores and corresponding heterogeneities.

A major concern when using DEX for FNB is the increased risk of hemodynamic instability, which usually manifests as bradycardia and hypotension. Pooled data in the current review also showed that use of DEX for FNB increases hypotension when compared with sham control. The underlying mechanisms may include declined plasma levels of norepinephrine and epinephrine caused by the use of DEX [49, 50]. However, it should be noted that, among those 4 studies which reported the incidence of hypotension, statistical significance was only found

in one study [22]. Similar phenomenon was also reported in some previous articles [16, 43]. On the other hand, combined use of DEX with local anesthetics for FNB tended to decrease PONV although not significantly so; possibly due to decreased opioid consumption.

Some limitations should not be neglected in this study. Firstly, significant heterogeneity existed when analyzing the analgesic indicators including postoperative pain scores, analgesic duration, and morphine consumption. In addition to FNB, NSAIDs, tramadol and/or PCIA were also applied for multimodal analgesia. It is difficult to eliminate the influence of these factors when performing subgroup analysis or sensitivity analysis. Secondly, only four or five studies were eligible for data synthesis for each analgesic outcome. Therefore, it was hard to conduct a meta-regression analysis to find more potential origins of heterogeneity or to draw funnel plots to evaluate the publish bias. Lastly, the effects of DEX used for FNB were investigated only in two surgical procedures (knee arthroplasty and knee arthroscopy), it is difficult to generalize our results to other clinical applicability.

## Conclusion

In conclusion, DEX when used as an adjuvant to local anesthetics for FNB improves analgesia, prolongs analgesic duration and reduces supplemental opioid requirements in patients following lower extremity surgery. However, DEX use increases the risk of postoperative hypotension which should be taken into considerations.

## Supporting information

**S1 Checklist. PRISMA 2009 checklist of the present study.**
(DOCX)

**S1 File. Search strategy of the present study.**
(DOCX)

## Author Contributions

**Conceptualization:** Zi-Fang Zhao, Dong-Xin Wang.

**Data curation:** Zi-Fang Zhao, Lei Du.

**Formal analysis:** Zi-Fang Zhao, Lei Du, Dong-Xin Wang.

**Investigation:** Zi-Fang Zhao, Lei Du.

**Methodology:** Zi-Fang Zhao, Dong-Xin Wang.

**Project administration:** Zi-Fang Zhao, Lei Du.

**Software:** Zi-Fang Zhao, Lei Du.

**Supervision:** Dong-Xin Wang.

**Validation:** Dong-Xin Wang.

**Writing – original draft:** Zi-Fang Zhao, Lei Du.

**Writing – review & editing:** Dong-Xin Wang.

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
