## [Decision Letter · Decision Letter 0]

19 Aug 2020

PONE-D-20-23215

Effects of dexmedetomidine as a perineural adjuvant for femoral nerve block: a systematic review and meta-analysis

PLOS ONE

Dear Dr. Dong-Xin Wang ,

Thank you for submitting your manuscript to PLOS ONE. After careful consideration, we feel that it has merit but does not fully meet PLOS ONE’s publication criteria as it currently stands. Therefore, we invite you to submit a revised version of the manuscript that addresses the points raised during the review process.

We look forward to receiving your revised manuscript.

Kind regards,

Laura Pasin

Academic Editor

PLOS ONE

Journal Requirements:

Reviewers' comments:

Reviewer's Responses to Questions

**Comments to the Author**

1. Is the manuscript technically sound, and do the data support the conclusions?

Reviewer #1: Partly

Reviewer #2: Yes

2. Has the statistical analysis been performed appropriately and rigorously? 

Reviewer #1: Yes

Reviewer #2: Yes

3. Have the authors made all data underlying the findings in their manuscript fully available?

Reviewer #1: Yes

Reviewer #2: Yes

4. Is the manuscript presented in an intelligible fashion and written in standard English?

Reviewer #1: Yes

Reviewer #2: Yes

5. Review Comments to the Author

Reviewer #1: Dear editor,

Dear authors,

I read with pleasure the paper entitled "Effects of dexmedetomidine as a perineural adjuvant for femoral nerve block: a systematic review and meta-analysis" In their paper Zi-Fang Zhao and colleagues investigated the the benefit and effectiveness of dexmedetomidine as adjuvants to local anesthetics for femoral nerve block.

Below I am reporting some concerns:

Was this meta-analysis registered prospectically? Please provide registration number.

Search strategy

Please present the search strategy for each database as supplementary material

"Subsequently, we identified the possibly included trials by carefully reading the full text". Do you mean reading the references in the full text?

Inclusion and exclusion criteria:

Please present inclusion and exclusion criteria as PICOS.

Bias

You used Cochrane’s risk-of-bias tool, however it is an out of date tool, please use ROB2

Do you contact authors for missing data? If not please state that in method section

Results

One study was excluded because the effects could not be estimated with RevMan software. What means?

Please use GRADE to assess quality of the evidence

Please provide a table with sensitivity analysis

Discussion

"increases hypotension and lowers blood pressure when compared with sham control." aren't they synonim synonyms?

"However, it should eb noted that" typo

Reviewer #2: The authors present a meta-analysis analyzing the effects related to the use of Dex during FNB.

Peripheral nerve blocks and opioid-sparing techniques are topics interesting and under scientific debate.

The limits of the analysis are well described in the discussion.

6. PLOS authors have the option to publish the peer review history of their article (what does this mean?). If published, this will include your full peer review and any attached files.

Reviewer #1: **Yes: **Alessandro De Cassai

Reviewer #2: No

---

## [Author Response · Author response to Decision Letter 0]

15 Sep 2020

Reviewer #1:

1. Was this meta-analysis registered prospectically? Please provide registration number.

Response: Thanks for your sincere comment. We failed to register this meta-analysis prospectively and this was a limit.

2. Please present the search strategy for each database as supplementary material.

Response: Thanks for pointing out this. We have added the search strategies for all database and submitted in “Revised Supplementary 2”.

3. "Subsequently, we identified the possibly included trials by carefully reading the full text". Do you mean reading the references in the full text?

Response: Thanks for your careful reading and prudent attitude. We revised the sentence as below in order to clarify the procedure: “Subsequently, the identified articles were screened by reading the title and retrieved abstracts. Full text of selected articles was carefully read for possible inclusion. We also checked the reference lists of selected articles to avoid the omission of any eligible trials (page 4, line 14)”.

4. Please present inclusion and exclusion criteria as PICOS.

Response: As suggested, we have revised the inclusion and exclusion criteria accordingly.

5. You used Cochrane’s risk-of-bias tool, however it is an out of date tool, please use ROB2.

Response: Thanks for your advice. We have provided a “Revised Figure 2” which presents the details of risk of bias by using RevMan software.

6. Do you contact authors for missing data? If not please state that in method section.

Response: Thanks for pointing out this. We clarified this in the revised manuscript: “For studies that reported missing or insufficient data, we attempted to contact authors to acquire data. When feasible, data were extrapolated from figures or tables (page 5, line 17)”.

7. One study was excluded because the effects could not be estimated with RevMan software. What means?

Response: Thanks for careful reading. We revised the sentence in order to clarify the meaning: “One study was excluded because of limited data and the effects could not be estimated with RevMan software (page 11, line 9)”.

8. Please use GRADE to assess quality of the evidence.

Response: Many thanks for your suggestion. We added a paragraph (page 12, line 4) in the main text and Table 2 (page 13) to describe the GRADE evaluations in the revised manuscript.

9. Please provide a table with sensitivity analysis.

Response: Thanks for your kind suggestion. In the present meta-analysis, statistical analysis was performed with RevMan software. Sensitivity analysis was performed by excluding the study with a potentially high risk of bias when doing analysis with the software; no notable changes were found in all three timepoints after excluding any of these studies. The process cannot be exported as images or tables. We reported these findings in the Result section as below.

In the “Pain scores in resting state” section, “Sensitivity analysis was conducted to detect the origin of heterogeneity, but no notable changes were found in all three timepoints after excluding any of these studies (page 9, line 23).”

In the “Duration of analgesic effects” section, “Sensitivity analysis was performed, but the recalculated MD and heterogeneity after excluding any of the trials showed no significant changes (page 11, line 2).”

In the “Pain scores in active state” section, we reported that “The effect of DEX in reducing active pain score was not altered after divided into single-shot FNB [19, 21, 23] and continuous FNB subgroups [25-27] (page 10, line 14).”

10. "increases hypotension and lowers blood pressure when compared with sham control." aren't they synonim synonyms?

Response: Thanks for pointing out this. We have corrected the sentence: “Pooled data in the current review also showed that use of DEX for FNB increases hypotension when compared with sham control (page 17, line 5)”.

11. "However, it should eb noted that" typo.

Response: We have made correction (page 17, line 7).

Reviewer #2: 

1. The authors present a meta-analysis analyzing the effects related to the use of Dex during FNB. Peripheral nerve blocks and opioid-sparing techniques are topics interesting and under scientific debate. The limits of the analysis are well described in the discussion.

Response: Thank you very much.

---

## [Editor Report · Decision Letter 1]

29 Sep 2020

Effects of dexmedetomidine as a perineural adjuvant for femoral nerve block: a systematic review and meta-analysis

PONE-D-20-23215R1

Dear Dr. Wang,

We’re pleased to inform you that your manuscript has been judged scientifically suitable for publication and will be formally accepted for publication once it meets all outstanding technical requirements.

Kind regards,

Laura Pasin

Academic Editor

PLOS ONE
---

## [Editor Report · Acceptance letter]

5 Oct 2020

PONE-D-20-23215R1 

Effects of dexmedetomidine as a perineural adjuvant for femoral nerve block: a systematic review and meta-analysis 

Dear Dr. Wang:

I'm pleased to inform you that your manuscript has been deemed suitable for publication in PLOS ONE. Congratulations! Your manuscript is now with our production department. 

Kind regards, 

on behalf of

Dr. Laura Pasin 

Academic Editor

PLOS ONE